# Insulin and Insulin Resistance in Alzheimer’s Disease

**DOI:** 10.3390/ijms22189987

**Published:** 2021-09-15

**Authors:** Aleksandra Sędzikowska, Leszek Szablewski

**Affiliations:** Chair and Department of General Biology and Parasitology, Medical University of Warsaw, Chalubinskiego 5, 02-004 Warsaw, Poland; aleksandra.sedzikowska@wum.edu.pl

**Keywords:** Alzheimer’s disease, central nervous system, insulin, insulin-like growth factor, insulin resistance, type 2 diabetes mellitus

## Abstract

Insulin plays a range of roles as an anabolic hormone in peripheral tissues. It regulates glucose metabolism, stimulates glucose transport into cells and suppresses hepatic glucose production. Insulin influences cell growth, differentiation and protein synthesis, and inhibits catabolic processes such as glycolysis, lipolysis and proteolysis. Insulin and insulin-like growth factor-1 receptors are expressed on all cell types in the central nervous system. Widespread distribution in the brain confirms that insulin signaling plays important and diverse roles in this organ. Insulin is known to regulate glucose metabolism, support cognition, enhance the outgrowth of neurons, modulate the release and uptake of catecholamine, and regulate the expression and localization of gamma-aminobutyric acid (GABA). Insulin is also able to freely cross the blood–brain barrier from the circulation. In addition, changes in insulin signaling, caused inter alia insulin resistance, may accelerate brain aging, and affect plasticity and possibly neurodegeneration. There are two significant insulin signal transduction pathways: the PBK/AKT pathway which is responsible for metabolic effects, and the MAPK pathway which influences cell growth, survival and gene expression. The aim of this study is to describe the role played by insulin in the CNS, in both healthy people and those with pathologies such as insulin resistance and Alzheimer’s disease.

## 1. Introduction

Human insulin is a 51-amino acid peptide hormone consisting of two chains, synthesized in the β-cells of the islets of Langerhans in the pancreas. The hormone protects against protracted elevations of circulating blood glucose levels, i.e., hyperglycemia; it is secreted into the peripheral blood with the aim of counteracting a rise in exogenous or endogenous glucose.

In the peripheral tissues, insulin facilitates glucose utilization, suppresses hepatic glucose production and promotes the transport of glucose into cells by translocation of glucose transporters, such as GLUT 4, from the intracellular compartment into the plasma membrane. Insulin also influences cell growth, differentiation and protein synthesis. As an anabolic hormone, it promotes the uptake of fatty acids and amino acids and enables energy storage. However, it also inhibits processes, such as glycolysis, lipolysis and proteolysis. 

It is present at much lower levels in the cerebrospinal fluid (CSF) than in the plasma [1]; however, correlations have nevertheless been observed between the two levels, suggesting that most insulin in the brain is derived from circulating pancreatic insulin [2]. Insulin can cross the blood–brain barrier (BBB) from the circulation to the brain via the capillary endothelial cells of the BBB by a selective, saturable, receptor-dependent mechanism [3]. When in the brain, insulin binds to the insulin receptor (IR) to form an insulin–insulin receptor complex, which is transported by transcytosis through the brain to the endothelial cells. This transport can be modulated by several factors, such as a high-fat diet, astrocyte stimulation [4], obesity, inflammation, diabetes mellitus, and circulating triglyceride levels [5]. In addition, animal studies have found that CSF levels are lower during fasting, and increase after meals [6,7]. 

Insulin resistance may cause disruption of the BBB, changing its permeability. This impairment causes cerebrovascular dysfunction, resulting in deficits in synaptic plasticity and cognition.

Some regions of the brain, such as the hypothalamus and choroid plexus, may serve as sites of more rapid entry for peripheral insulin into the central nervous system (CNS) [8]. It has also been proposed that some insulin synthesis may take place de novo in the brain: insulin mRNA has been identified in selected brain regions in rats and mice [5,9], and the presence of C-peptide, a by-product of local insulin synthesis, has been described in human CSF [10]. In addition, insulin mRNA has been reported in post-mortem studies of human brain tissue, especially in the hippocampus and hypothalamus [2]. 

Insulin production in the brain remains controversial. It remains unsettled as to whether insulin is actually produced in the brain of higher-order organisms. In chordates, insulin mRNA is found in the nervous system of tunicates, as well as in the brain and pituitary gland of fish. Insulin has been described in several regions of the brain in mammals, but its secretion has also been reported in cultured astrocytes. In mice and humans, it may also be produced by choroid plexus, but its release is modulated, not by glucose, but by serotonin.

## 2. The Role of Insulin in CNS

All cell types in the brain express insulin receptors; however, their expression differs between particular regions of the brain. Animal studies indicate the highest densities of IRs in the CA1, CA3 olfactory bulb, entorhinal cortex and hypothalamus and the dentate gyrus regions of the hippocampus, as well as the cerebral cortex, striatum and cerebellum [11,12,13,14,15]. Insulin receptors have also been detected in human brains [16,17]. The widespread distribution of these receptors in the brain suggests that insulin signaling plays a range of important roles therein [2]. On the other hand, changes of insulin signaling in CNS may accelerate brain aging, affect plasticity and may be involved in the process of neurodegeneration [18]. 

### Insulin Signaling in the Brain

Insulin and insulin-like growth factor 1 (IGF-1) exert biological effects through the insulin receptor and the IGF-1 receptor (IGF-1R), both of which are widely distributed throughout the brain [19]. Animal studies indicate that IRs are highly expressed in the olfactory bulb, hippocampus, neocortex, hypothalamus and cerebellum. In mouse brains, IGF-1R is highly expressed in the hippocampus, neocortex and thalamus, and at a lower level in the hypothalamus, cerebellum, olfactory bulb, midbrain and brainstem [20,21]. Both receptors can heterodimerize in the brain and partially transactivate their signaling [22]. IR can be further subdivided into two isoforms: IR-A, found in the adult nervous system, and IR-B, expressed mainly in adipose tissue, liver and skeletal muscle [5,23], as well as in astrocytes, albeit at a lower level [24]. IR-A is the short form, lacking 12 amino acids within the C-terminus of the α-subunit [25]. Of note, IGF-1 and IGF-2 can also bind at the IR [5,26]. Insulin receptors are expressed in both neurons and glia. 

Many molecules are involved in insulin signaling cascades, known to play key roles in brain functions. Two of these are insulin signal transduction pathways: one, PI3-K/AKT (phosphatidylinositol 3-kinase/protein kinase B, PKB), is responsible for metabolism and lipid and protein synthesis, while the other, MAPK (mitogen-activated protein kinase) influences cell growth, survival and gene expression [26,27]. 

Insulin signaling begins with the binding of insulin to a tyrosine kinase receptor, which acts as a transmembrane insulin receptor: a hetero-tetrameric membrane glycoprotein composed of two α- and two β-subunits. IGF-1 also binds and activates IRs. Thus, IR and IGF-1R initiate many of the same trophic actions [28]. Once insulin is bound to the extracellular α-subunit of IR, it induces the dimerization of the intracellular β-subunit. The α-subunit also promotes autophosphorylation of tyrosine residues on the β-subunit by activation of intrinsic tyrosine kinases. The auto-phosphorylated β-subunit then phosphorylates tyrosine residues on a group of adaptor proteins belonging to IRS (insulin receptor substrate) families 1 through 4 (IRS1-IRS4) [29]. Of these, the best characterized are IRS1 and IRS2, which are the most widely-distributed forms. In the brain, IRS1 is expressed in the cerebral cortex, and IRs in the hypothalamus [2]. The dephosphorylation of tyrosine residues in IR and IRS1 may also regulate downstream insulin signaling via protein tyrosine phosphatase 1B (PTB1B) [30]. 

IRSs differentiate insulin stimulation into various pathways [31] and transmit intracellular signals that mediate different processes. The activation of the insulin–IR–IRS–PI3-K–AKT pathway regulates the phosphorylation of many intracellular proteins, such as serine/threonine-protein kinase mTOR (mammalian target of rapamycin), glycogen synthase kinase 3β (GSK3β), cAMP-responsive element-binding protein (CREB), filamin A and nitric oxide synthetases. This insulin signaling pathway is involved in several processes, such as DNA replication, cell cycle activity, protein synthesis, cell survival, metabolism, angiogenesis, potassium uptake, modification of lipid and autophagy [2]. It stimulates the translocation of GLUT 4 from the intracellular space to the plasma membrane, increasing glycolytic metabolism [32]. Stimulation of this pathway preserves mitochondrial membrane integrity [33] and inhibits the production of free radicals that damage mitochondrial DNA and pro-apoptosis mechanisms [34]. 

As mentioned above, the second main insulin signaling pathway is insulin–IR–IRS–Raf/Ras/MAPK. This pathway is activated by the growth factor receptor-bound protein-2 (Grb-2) binding to Tyr-phosphorylated Src homology collagen (Shc) or IRS via its Src homology 2 (SH2) domain. The MAPK signaling pathway includes extracellular signal-regulated kinases 1 and 2 (ERK1 and ERK2), p38 and c-Jun-N-terminal kinases (JNKs). This pathway controls various transcription factors and elements such as CREB, and the proto-oncogenes cMyc (MYC) and c-Fos (FOS). It also regulates the transcription, translation and post-translational modification of various proteins, such as growth factors, receptor genes and matrix-modifying proteins [2]. This signal is involved in the expression of genes associated with glucose metabolism, GLUT 3 biosynthesis and induction of mitosis in cells [35] (Figure 1).

## 3. The Role of Insulin in the Brain

Insulin receptors are widely distributed throughout the brain. As mentioned earlier, their highest density has been detected in the olfactory bulb, hypothalamus, hippocampus, cerebral cortex and cerebellum [36]. Most insulin receptors are localized on neurons; they are highly expressed in the presynaptic axon terminal of synapses and are a component of the postsynaptic density (PSD) [2]. As such, the synapse appears to be an important site of specialized insulin signaling in the brain [37], with the most commonly expressed form being the short isoform A (IR-A).

### 3.1. Insulin and Brain Glucose Metabolism

Most glucose uptake in neurons takes place via GLUT 3, which is co-expressed with a number of insulin-regulated GLUT proteins, such as GLUT 4 and GLUT 8. These glucose transporters are localized within specific brain regions, such as the basal forebrain, hippocampus, amygdala, sensorimotor cortex, hypothalamus, pituitary, and to a lesser degree, the cerebral cortex and cerebellum [2,38]. Insulin stimulates the expression of GLUT 4 and its translocation from the cytosol to the plasma membrane, facilitating the uptake and utilization of glucose. The regulation of neuronal metabolism and the generation of energy, necessary for cognition and memory, are dependent on stimulation of GLUT by insulin [39]. Induction of GLUT 4 translocation to the neuron cell membrane via the AKT pathway improves insulin transport into neurons during periods of high metabolic demand, such as during learning. This result suggests that the deregulation of insulin-dependent glucose transport in several brain regions may cause cognitive impairment [40,41]. Animal studies indicate that translocation of GLUT 4 to the plasma membrane in rat hippocampus following insulin stimulation increases glycolysis and enhances spatial memory [42]. In astrocytes, stimulation of GLUT 4 promotes glucose uptake and glycogen accumulation [43]; however, the role of insulin in astrocyte functions remains unclear [44]. GLUT 8, another glucose-dependent glucose transporter, is expressed in several areas of the brain, in particular in the hippocampus. It is involved in glucose homeostasis in neurons [45,46]. 

The activity of insulin in the brain also influences the peripheral metabolism [18]. Insulin signaling in the CNS regulates the metabolic pathways in organs and tissues, such as liver and adipose tissue. These effects are caused by the action of hormones in the hypothalamus [2]. The metabolic effects of brain insulin, such as suppression of hepatic glucose production, lipolysis in adipose tissue, catabolism of branched-chain amino acids in the liver and hepatic secretion of triglyceride have been attributed to the modulation of vagal and/or sympathetic efferent fibers [2].

### 3.2. Insulin and Cognition

The high concentration of insulin receptors in the hippocampus, entorhinal cortex and frontal cortex, i.e., the brain regions involved in learning and memory, suggests that insulin plays a key role in these processes. In addition, changes in insulin receptors have been observed in the hippocampus secondary to spatial learning, as have improvements in memory after insulin administration. These effects were observed in both animal models and human studies; however, the action of insulin on cognition is not definite [38]. The results of animal studies suggest that insulin contributes to changes in hippocampal synaptic plasticity by favoring long-term potentiation (LTP) [47] and long-term depression (LTD) [48]. These molecular mechanisms are involved in hippocampal-dependent learning and memory [49]. One such mechanism, regulated by insulin, is based on the expression of *N*-methyl-D-aspartate (NMDA) receptor in the membrane [50]; this expression is regulated by the insulin-activated ERK1/2 [51] or PI3-K [52] pathways. Insulin signaling also plays an important role in synaptic remodeling, which is essential for neuronal plasticity [37]. Other observations suggest that intact insulin signaling is needed for memory formation [53]. 

The expression and function of the insulin receptor in specific brain regions may be modified by the process of learning. Observations performed in animal studies indicate that spatial memory training upregulates IR mRNA in the hippocampal CA1 region and dental gyrus, resulting in an increased accumulation of insulin receptors within the hippocampus. In water maze-trained rats, training increases tyrosine phosphorylation of the insulin receptor stimulated by insulin [54]. The obtained results suggest that learning may influence the concentration of insulin receptors and insulin signaling in various regions of the brain including the hippocampus. In addition, insulin may play an important role in learning and memory, influencing, for example, the localization of IRs in the hippocampus [38].

The expression and function of the insulin receptor in specific brain regions may be modified by the process of learning. Observations performed in animal studies indicate that spatial memory training upregulates IR mRNA in the hippocampal CA1 region and dental gyrus, resulting in an increased accumulation of insulin receptors within the hippocampus. In water maze-trained rats, training increases tyrosine phosphorylation of the insulin receptor stimulated by insulin [54]. The obtained results suggest that learning may influence the concentration of insulin receptors and insulin signaling in various regions of the brain including the hippocampus. In addition, insulin may play an important role in learning and memory, influencing, for example, the localization of IRs in the hippocampus [38].

A high density of IRs in the limbic cortical and subcortical regions may also affect other aspects of mental functioning, such as mood, reward and motivation [2]. 

Insulin signaling in the CNS also affects emotional regulation. Rats subjected to lentivirus-mediated downregulation of hypothalamic insulin receptors demonstrated depression- and anxiety-like behaviors [55]. In addition, insulin administration was found to enhance object-memory and yield anxiolytic effects on behavior in mice [56], and chronic brain stimulation by intranasal insulin administration was found to improve memory in humans [56]. 

These observations suggest that the hippocampal insulin receptor is a key target in physiological cognitive processes [57]. 

The aforementioned observations suggest that the hippocampal insulin receptor is a key target in physiological cognitive processes [57]. 

### 3.3. The Effects of Insulin in Neurons

As mentioned earlier, insulin receptors are strongly expressed in both the presynaptic axon terminal and the postsynaptic compartments of synapses. The hormone also exerts a number of effects in neurons via the AKT and MAPK signaling pathways [58,59]: it is believed to enhance the outgrowth of neurons, modulate the release and uptake of catecholamine, and regulate the expression and localization of γ-aminobutyric acid (GABA), a major inhibitory neurotransmitter in mammalian brain synapses. GABA is known to play important roles in sleep, learning and memory and the activity of the reproductive system [60] It also has a significant influence on the regulation of food intake, body weight, plasticity and neuronal activity in the frontal cortex [61]. Insulin also regulates the expression of NMDA and α-amino-3-hydroxy-5-methyl-4-isoxazole propionic acid (AMPA) receptors [2] and it modulates activity-dependent synaptic plasticity, i.e., long-term potentiation and long-term depression via NMDA receptor signaling and the AKT pathway [48]. 

Insulin plays a key role in the development and maintenance of the excitatory synapses and promotes the formation of the dendritic spine [62,63]. The regulation of AKT and GSK3β by insulin is an important part of the modulation of the balance between LTP and LTD [64]. Insulin also promotes neuronal survival by inhibiting apoptosis via stimulation of the AKT signaling pathway [65]; it also stimulates the phosphorylation of the FOXO transcription factor, which is involved in the expression of pro-apoptotic mediators, resulting in cell death. Intrahippocampal microinjection of insulin also has a dose-dependent effect on cognitive function in animal studies; high doses of insulin were found to significantly ameliorate spatial learning and memory in the Morris Water Maze (MWM) test, whereas low doses of insulin cause reduction of cognitive performance [66].

### 3.4. Effects of Insulin in Glial Cells

The principal homeostatic cells of grey matter in the human brain are astrocytes [67]. Glucose is transported into astrocytes by GLUT 1, which can also transport lactate to neurons as an alternative source of energy during hypoglycemia: a process known as the astrocyte-neuron lactate shuttle [68,69]. Astrocytes express IR, IRS1 and IRS2 and the downstream signaling molecules AKT and MAPK. These signaling pathways are stimulated by insulin and IGF-1. Chronically high levels of insulin result in the downregulation of insulin receptors in glial cells, without any apparent change in neuronal IR expression. It is important to note that astrocytes also play a role in the inflammatory responses in the brain: insulin modulates the secretion of inflammatory cytokines by astrocytes in response to inflammatory factors [70]. 

The proliferation of oligodendrocytes, survival, differentiation and myelination is also influenced by AKT signaling, with the activation of AKT signaling pathway by IGF stimulating differentiation and axonal ensheathment [71].

Insulin also modulates the inflammatory responses of microglia in a complex manner [70], enhancing the secretion of certain inflammatory cytokines and inhibiting the secretion of others in vivo.

### 3.5. Insulin and Hippocampal Adult Neurogenesis

Neurons are generated in the newborn hippocampus and persist throughout adulthood [72]. In all mammals, adult neurogenesis occurs in the sub-granular zone of the hippocampus [73], where neural stem cells (NSCs) proliferate and differentiate to generate new neurons [74]. Hippocampal neurogenesis plays an important role in learning and memory, and its impairment in neurodegenerative disorders is associated with cognitive dysfunction [75]. Brain development and control of neurogenic niches is strictly dependent on insulin, which acts as a trophic factor. The activation of the insulin/IGF-1 pathway regulates the exit of neuroblasts from quiescence [76,77]. In addition, insulin and IGF-1 promote neurogenesis by modulating NSC proliferation, differentiation and survival [78,79]. Chronic hyper-activation of insulin/IGF-1 signaling pathways has also been found to result in premature depletion of the NSC reservoir [80]. Based on the timing and the duration of stimulation, insulin may have either trophic or detrimental effects on the neural stem niche [44].

## 4. Insulin Resistance and Alzheimer’s Disease

In type 2 diabetes, insulin resistance is defined as “reduced sensitivity in body tissues to the action of insulin” [81]. Similarly, brain insulin resistance may be defined as the failure of brain cells to respond to insulin [82]. Lack of response to insulin may be due to various causes, including downregulation of insulin receptors, the inability of IR to bind insulin or an impairment of the insulin cascade [83]. 

At the cellular level, insulin resistance may be understood as the impairment of neuroplasticity or the release of neurotransmitters in neurons. However, in both the brain or periphery, such resistance may result in a decreased ability to regulate the metabolism or impaired cognition and mood [2]. The peripheral metabolic disorders observed in type 2 diabetes are similar to the abnormalities present in brains of patients with Alzheimer’s disease (AD). Therefore, it has been suggested that AD may represent a form of type 2 diabetes mellitus associated with the brain [84], and may be regarded as type 3 diabetes [85].

Parkinson’s disease (PD) is a neurodegenerative disorder characterized by the accumulation of alpha-synuclein in neurons forming Lewy bodies. In the pathogenesis of PD, a potential involvement of abnormal insulin/IGF-1 signaling has been suggested. Parkinson’s disease and dementia with Lewy bodies (DLB) frequently overlap with Alzheimer’s disease, which is linked to brain impairments in insulin, insulin-like growth factor, and neurotrophin signaling. Decreased expression of insulin, IGF-II, and receptors of insulin is observed, as well as IGF-I, and IGF-II in PD and DLB frontal white matter, amygdala, and in DLB frontal cortex. Signaling of IGF-I, IGF-II, and neurotrophin are more impaired in DLB than PD, corresponding with DLB’s more pronounced neurodegeneration, oxidative stress, and alpha-synuclein accumulation, causing PD/DLB associated abnormalities in central nervous system neurons, and therefore may contribute to their molecular pathogenesis. Antidiabetic drugs that facilitate insulin/IGF-1 signaling have beneficial effects on motor symptoms and cognition in PD patients.

### 4.1. Insulin Resistance and Tau Phosphorylation

The Tau protein was first discovered in 1975, and may play a role in the dysregulation of insulin signaling and pathogenic alterations in the brain [86]. Indeed, it has been identified as the main component of neurofibrillary tangles in the brain of patients with AD [87]. The human brain has been found to harbor six isoforms of Tau generated by alternative splicing [88] and it has been classified as a microtubule-associated protein (MAP) [89]. 

The Tau protein is involved in the assembly and stability of microtubules that play a role in several cellular processes, such as cell morphogenesis, cell division and intracellular trafficking. It also influences the synapses and nuclei of neurons [90], from where it is also released into the extracellular space [91]. However, its extracellular function remains unknown [92]. The activity of Tau protein is modulated by phosphorylation, and the molecule itself contains more than 85 potential phosphorylated or phosphorylable serine, threonine and tyrosine sites [93]. In addition, both the expression of the *Tau* gene and the phosphorylation of Tau protein itself are regulated by insulin and IGF stimulation [94].

In brains characterized by AD, Tau protein demonstrates three times as much hyperphosphorylation than in normal brains [83]. Indeed, out of 85 phosphorylable residues, more than 40 phosphorylation sites have been identified in Tau from the brains of AD patients, and 28 sites are exclusively phosphorylated [95]. In vitro studies indicate that more than 30 kinases are known to regulate the process of Tau phosphorylation, and imbalances between these kinases and phosphatases are believed to result in hyperphosphorylation. 

GSK3-β phosphorylates Tau at more than 30 sites, and hence may play a key role in the development of AD and neurofibrillary tangle (NFT) [96]. In addition, several kinases and phosphatases, such as GSK3-β, AMPK, ERK, c-Jun N-terminal kinases (JNK), protein phosphatase 1 (PP1) and PP2 appear to be involved in the regulation of insulin signaling [49,95]. Furthermore, impaired insulin or IGF-1 signaling may enhance Tau phosphorylation by inhibition of PI3-K/AKT and increased GSK3-β activation [94]; inhibition of insulin/IGF-1 signaling also inhibits the Wnt signaling pathway [97], which negatively regulates GSK3-β. Interestingly, GSK3-β can also be activated by oxidative stress, which is a consequence of insulin resistance. Finally, Tau hyperphosphorylation can also be enhanced by the downregulation of Tau *O*-GlcNAcylation caused by decreased glucose metabolism due to insulin resistance [98]. 

Hyperphosphorylation forces conformational changes in Tau that impair its ability to bind to microtubules; as a result, misfolded Tau monomers cannot be transported into axons to accumulate, oligomerize and aggregate in neuronal perikaria. During the pathological aggregation process, Tau protein accumulates in domains with a beta sheet conformation and form filaments. Tau aggregates may become deposited in NFT, resulting in the manifestation of a group of diseases called tauopathies. Hyperphosphorylated Tau reduces the potential of normal Tau to influence microtubules by preventing it from binding to tubulin. This has been demonstrated to affect cell morphology and growth, and the transport of organelles mediated by microtubule-dependent motor proteins [83]. Intraneuronal accumulation of hyperphosphorylated Tau enhances oxidative stress and triggers pathologies such as increased apoptosis, mitochondrial dysfunction and necrosis [99]. 

Animal studies indicate that Tau protein can regulate brain insulin signaling [100], with Tau deletion impairing the hippocampal response to insulin. Further impairment is caused by changes in the influence of IRS-1 and phosphatase and tensin homologue on chromosome 10 (PTEN), which is a negative regulator of the PI3-K/AKT pathway. Tau knockout mice exhibit an impaired hypothalamus, which is associated with changes in energy metabolism. The authors conclude that pathophysiological loss of Tau function may be associated with brain insulin resistance, and that this plays a key role in cognitive and metabolic impairments in AD patients [100]. Elsewhere, it has been observed that in cases of AD and several of the most prevalent human tauopathies, insulin accumulates as oligomers in hyperphosphorylated Tau-bearing neurons. The accumulation of insulin in neurons is directly related to the level of Tau hyperphosphorylation and follows the progression of tauopathy; in addition, the accumulation of insulin is related to insulin resistance and reduction in IR level [101].

As the Alzheimer’s disease symptoms progress, Tau pathology is observed first in the brainstem and entorhinal cortex, and later in the hippocampus [102]. It is important to note that Tau is necessary for normal brain function, as Tau deletion is also associated with the accumulation of iron in the brain, resulting in conditions such as Parkinson’s disease [103] (Figure 2). 

### 4.2. Insulin Resistance and Amyloid-β Pathology

Amyloid precursor protein (APP) is a transmembrane protein expressed in many tissues including the CNS, and is concentrated in the synapses of neurons. It functions as a cell surface receptor and has been implicated as a regulator of synapse formation, neural plasticity, antimicrobial activity and iron export. APP is processed by two different pathways: 90% by the non-amyloidogenic (nonplaque-forming or secretory) pathway, and the remaining 10% by the amyloidogenic pathway. Insulin promotes the non-amyloidogenic processing of APP by regulating its phosphorylation. Therefore, impairment of insulin signaling may increase the accumulation of pathological amyloid-β (Aβ) [104]. Insulin and amyloid-β are substrates for insulin degrading enzyme (IDE). As insulin increases the level of IDE, a defect in insulin signaling leads to reduced amyloid-β degradation [105]. 

In the non-amyloidogenic pathway, APP is segregated via α-secretase, thus releasing a C-terminal fragment α (CTFα, C83) and a soluble N-terminal fragment α (sAPPα). CTFα is then cleaved by γ-secretase to obtain a smaller C-terminal fragment (C3) [83,95]. In the amyloidogenic (plaque forming pathway), taking place within the endosome with an acidic environment, β-secretase is induced to cleave APP into a smaller N-terminal (sAPPβ) fragment and a longer C-terminal fragment containing the full amyloidogenic aminoacidic sequence (CTFβ, C99). Subsequent cleavage of CTFβ by γ-secretase results in the formation of an APP intracellular cytoplasmic domain (AICD) and Aβ fragments, both of which are released into the extracellular environment. In the extracellular environment, the Aβ fragments rapidly form oligomers, prefibrillar aggregates and fibrils that will form β-amyloid plaques [83,95]. 

Aβ peptide chain contains 38 (Aβ_38_), 40 (Aβ_40_), and 42 (Aβ_42_) amino acids. Increased APP gene expression, together with altered proteolysis, results in the accumulation of 40 or 42 aa-length Aβ peptides. In early-onset AD, i.e., the familial form of AD, mutations in the APP, presenilin (PS)-1 and PS2 genes, and inheritance of the Apolipoprotein E ε4 (ApoE-ε4) allele change the role of γ-secretase, increasing the production of Aβ_42_ [106]. These mutations increase the synthesis and deposition of Aβ peptides in the brain. However, the causes of Aβ accumulation in sporadic AD are the subject of intense investigation [107]. Native APP facilitates the process of memory and learning via synapse activity and the formation of the dendritic spine [108]. In physiological conditions, Aβ is released to the extracellular space during neuronal activity, and its levels are controlled by local proteases [109]. Errors in the cleavage position may cause increased levels of the Aβ_42_ neurotoxic isoform. As mentioned above, Aβ peptides aggregate into oligomers, organize fibrils and form amyloid plaques. Aβ plaques block the signaling pathway and cell connections, which can lead to cell death. Intracerebral administration of neurotoxic Aβ causes a memory deficit in animals, similar to that observed in AD patients; such memory and learning deficits are associated with the disruption of synaptic plasticity caused by intraneuronal accumulation of Aβ [110]. It has been proposed that Aβ exerts its negative effects on neurons by activating the immune-inflammatory reaction in glial cells, resulting in the phagocytosis of neuronal and synaptic structures [111]. Thus, intraneuronal accumulation of Aβ is key factor in a number of synaptopathies. 

Insulin regulates the metabolism of APP, which in turn modulates the balance between the anabolism and catabolism of Aβ. Low levels of insulin, or the lack of its action, may increase NFT formation and result in oxidative damage to cells. In addition, low insulin levels result in elevated Aβ levels and form amyloid plaques in the brain. Insulin degrading enzyme has also been implicated in hyperinsulinemia and AD by dysregulating metabolic and neurological pathways [95]. IDE is a thiol zinc-metallo-endopeptidase which cleaves molecules, such as insulin, Aβ, glucagon and calcitonin, and is known to regulate the levels of insulin and Aβ in the brain. Studies indicate increased immunoreactivity surrounding senile plaques, and reduced expression of IDE in the hippocampus of patients with AD. IDE negatively regulates insulin signaling by catalyzing insulin degradation, and regulates extracellular Aβ levels by degrading Aβ. Animal studies based on these transgenic mice found that overexpression of IDE results in the development of hyperinsulinemia, glucose intolerance and increased levels of Aβ in the brain. Therefore, it is suggested that Aβ influences AD neurodegeneration by impairing insulin signaling and promoting insulin resistance [99]. 

In other animal studies on effects of Aβ oligomers on the brain, Aβ_42_ peptide was found to induce hepatic insulin resistance in APP_swe_/PS1E9 mice due to activation of Janus Kinase 2 (JAK2). As such, it is possible that the inhibition of Aβ_42_ peptide synthesis in the brain may be a strategy for the treatment of insulin resistance [112]. Administration of Aβ oligomers directly to the neurons in the hippocampus induce synaptic loss and neuronal dysfunction in rodents, which may be a cause of memory loss [113]. In addition, intracerebroventricular administration of Aβ oligomers causes behavioral changes and AD-like pathology in primates [114], while intracerebral injection of Aβ oligomers results in a range of pathological states, such as inflammation in the hypothalamus, as well as peripheral glucose intolerance, glucose resistance, inflammatory processes in adipose tissue, and disturbances in insulin-induced GLUT 4 translocation into the cell membranes of skeletal muscle [115]. An increased level of Aβ in the temporal lobe was observed in a monkey model of type 1 diabetes mellitus, with the greatest increase observed in the hippocampus; in addition, these parts of the brain demonstrated decreased levels of neprilysin (NEP), the Aβ-degrading enzyme, at the protein and mRNA levels [116]. 

Diet-induced brain insulin resistance has also been found to influence Aβ pathology in a mouse model of AD; the findings suggest that brain amyloid metabolism and amyloid pathology may be associated with diet-dependent insulin resistance. The authors conclude that, while the causal factors of insulin resistance, such as metabolic stress or inflammation due to a high-fat diet, may be directly involved in the acceleration of Aβ accumulation in the brain, insulin signaling does not appear to play a role [117].

Insulin influences the metabolism of the Aβ peptide by accelerating its trafficking to the plasma membrane from its point of origin in the trans-Golgi network. Insulin also increases the extracellular levels of Aβ by promoting its secretion and inhibiting its degradation by IDE. The effects of insulin on APP metabolism are influenced by downstream signaling through MAPK [118]. However, Aβ can also affect insulin signaling, either by competing with and inhibiting the binding of insulin to IR, or reducing its affinity [119]. Thus, Aβ accumulation may promote Tau hyperphosphorylation, resulting in tauopathies. Aβ may show adverse/neurodegenerative effects through its neurotoxic activities [120]. Intracellular Aβ interferes with the PI3-K activation of AKT, thus reducing signaling, while increasing GSK-3β activation and Tau hyperphosphorylation. Increased levels of GSK-3β promote APP processing and Aβ accumulation [121].

### 4.3. Insulin Resistance and Inflammation

The role of inflammation is well documented in the pathogenesis of AD. Aβ is believed to play a central role in the neuroinflammation hypothesis of AD, where its accumulation causes an increase of inflammatory cytokines, chemokines, and complement proteins which are synthesized and released by chronically-activated glia [122]. Inflammatory cytokines, such as interleukin-1 (IL-1), IL-6, tumor necrosis factor-α (TNF-α) and transforming growth factor-β (TGF-β), are commonly found to be elevated in the cerebrospinal fluid of AD patients [123]. 

Neuroinflammation is defined as “the presence of activated microglia and astrocytes which cause injury through expression and release of pro-inflammatory cytokines, chemokines, and complement increased generation of membrane fatty acids, eicosanoids, lipid peroxidation products, and reactive oxygen and reactive nitrogen species” [124]. In the brain, innate immunity is mediated by astrocytes and microglia. Microglial cells represent the immune system of the mammalian brain [109], and activation of these cells is a hallmark of central inflammation and, potentially, brain pathology [125]. 

Neuroinflammation is observed in many neurodegenerative diseases, and is an important propagator of the neurodegeneration observed in AD [126]; indeed, it may play one of the most important roles in the progression and prognosis of AD. Previous studies indicate that neuroinflammation causes neuronal injury and cholinergic dysfunction [127], as well as oxidative stress, increased production of reactive oxygen (ROS) and reactive nitrogen species (RNS). ROS and RNS may damage nerve terminals, causing synapse dysfunction [128]. Chronic inflammation exacerbates insulin resistance. A human study showed that peripheral insulin resistance may influence the pathology of AD via its action on neurodegeneration [129]. Peripheral insulin resistance increases the expression and activation of IDE and stimulates the accumulation of advanced glycation end products (AGEs) [107], i.e., glycated proteins and lipids formed through non-enzymatic glycosylation following exposure to glucose [130]. Adults with insulin resistance demonstrate higher AGE levels than healthy controls [131]. 

The accumulation of AGEs may have negative effects on tissues, and have been found in amyloid-containing senile plaques and Tau-containing neurofibrillary. Glycation is known to enhance Aβ aggregation. AGEs also stimulate vascular pathology in the brain [132]. AGEs target the receptor for advanced glycation end products (RAGE). RAGE is expressed in the CNS in neuronal cells, microglia, astrocytes and brain endothelial cells [133], and higher levels are observed in patients with AD and type 2 diabetes mellitus. RAGE binds AGE, causing upregulation of the nuclear factor-kappa beta (NF-κβ) transcription factor, an important mediator of inflammation [134]. RAGE also binds the Aβ peptide with high affinity; it can transport brain-derived Aβ across the BBB into the peripheral circulation [135] and back again [136], promoting neurodegenerative pathway activity. Binding to the Aβ peptide may promote the release of cytokines such as TNF-α and IL-6 from microglia, inducing subsequent neuronal damage due to the expression of macrophage-colony stimulating factor (M-CSF) [136,137]. 

The RAGE-ligand interaction may increase the expression of beta secretase-1, also known as beta-site APP cleaving enzyme 1 (BACE1), which promotes the generation of Aβ by the amyloidogenic processing of APP [138]. In addition, in AD, the interaction of RAGE with AGE-modified proteins or Aβ may cause complications in the vascular system, making it permeable to macromolecule invasion. In response to Aβ, RAGE mediates the migration of monocytes across human brain endothelial cells [139]. 

Macrophages are a major cellular component of neuroinflammation, and hence play a key role in the innate immune system. The inflammatory mediators released during inflammation impair insulin signaling via JNK pathway activation [140,141]. Insulin suppresses the activity of proinflammatory proteins, which downregulates the inflammatory response; it also induces the synthesis of postsynaptic density protein 95 (PSD-95) in Dendron’s and hippocampal area in states of inflammation/neuroinflammation, which is governed by the PB-K-AKT-mTOR signaling pathway [95]. A human study showed that a major risk factor of late onset AD is apolipoprotein E, which activates altered insulin-linked signaling networks in the brain. However, several other polypeptides and neuroinflammatory mediators are involved in insulin resistance in the brain in the pathogenesis and progression of AD [142]. 

### 4.4. Insulin Resistance and Oxidative Stress

Insulin resistance promotes oxidative stress through several routes such as the dysregulation of carbohydrate and lipid metabolism, increased GSK-3β activation, and the impairment of cell survival and anti-apoptotic signaling, energy balance and mitochondrial function [143]. Brain insulin resistance also impairs the expression of choline acetyltransferase and neurotrophin [144], and is associated with increased levels of phosphorylated Tau and Aβ_42_ [143]. Oxidative stress may also be caused by hypoxia and ischemia [143]. 

Oxidative stress increases the generation of ROS and RNS, which react with macromolecules such as lipids, proteins and nucleic acids, thus resulting in neurotoxic effects [145,146]. Oxidative stress is also believed to play a role in neurodegeneration, as damage due to oxidative stress occurs at the very early stages of neurodegeneration, and has been found to be associated with mitochondrial dysfunction in AD [147,148]. While oxidative stress is known to occur when cellular metabolic activity is greater than antioxidant capacity, it may also occur when the excessive amounts of ROS and RNS are produced, and these are too large to be removed [149]. 

Mitochondria have important physiological functions in the body, including oxidative respiration, energy metabolism and free radical production; they also delay aging and prevent neurodegenerative diseases. In addition, mitochondrial dysfunction has been associated with the pathogenesis of neurodegenerative diseases: impaired mitochondria produce less ATP, but generate more ROS, which may be a primary cause of the oxidative imbalance observed in AD. Enzymes involved in metabolic pathways such as glycolysis, the Krebs cycle and the respiratory chain become oxidized in the brains of AD patients; their resulting decreased activity inhibits the glucose metabolism in the brain resulting in reduced ATP synthesis, disrupted neuronal functioning, synapse loss and overall neurodegeneration [150,151]. The early stages of AD are characterized by increased activity of mitochondrial enzymes and oxidative stress; these dysfunctions have been found to occur prior to amyloid plaque accumulation in animal models of AD [152]. However, in vitro experiments suggest that the more neurotoxic Aβ oligomers can reduce the activity of cytochrome oxidase and increase ROS generation [153]. Due to the heterogeneity of the results obtained in previous studies [133], there is a need for further research to confirm whether Aβ oligomers initiate this process. 

### 4.5. Insulin Resistance and Cognitive Impairment 

Brain insulin deficiency and plasma insulin resistance may cause cognitive dysfunction. Insulin resistance promotes the development of cognitive dysfunction by hyperinsulinemia and impaired insulin signaling. In AD patients, the levels of insulin are low in the brain and CSF, but high in plasma, and this may be related to impaired signal transduction [154,155]. Insulin plays an important role as a long-term neuroprotectant, and its absence leads to neurodegeneration. Indeed, insulin resistance may act as an initial factor in cognitive dysfunction; unfortunately, the underlying mechanism is still unclear. Nevertheless, several mechanisms have been proposed for cognitive dysfunction, such as altered APP metabolism, elevated Tau protein concentration, inflammation in the brain, ApoE ε4 allele involvement and impaired hippocampal plasticity due to insulin resistance [156]. It is also suggested that brain insulin resistance may be an independent risk factor for cognitive dysfunction [157]. 

Higher brain functions such as learning and memory are associated with synaptic plasticity. Hippocampal synaptic plasticity, and thus learning and memory, are influenced by insulin; changes in the signal transduction pathway are known to impair cognitive function. As mentioned earlier, insulin modulates neurotransmission at the synapses; its presence has been associated with long-term depression and long-term potentiation, and is known to affect learning and memory through its influence on GABA receptors [157]. In animal models of AD, changes in insulin signaling cause memory impairment [42,158], while investigations of patients with prediabetes found that insulin resistance, rather than elevation of blood glucose, may cause cognitive decline, especially in the memory domain [159]. It was found that in cases of AD, the hippocampal formation and, to a lesser degree, the cerebral cortex exhibit reduced responses to insulin signaling, with the insulin–IR–IRS1–PI3-K and IGF-1–IR–IRS2–PI3-K signaling pathways being markedly impaired. It has been suggested that brain insulin resistance, closely associated with IRS-1 dysfunction, potentially triggered by the activity of Aβ oligomers, may promote cognitive decline, independent of the classic AD pathology [160]. 

Brain plasticity, i.e., its capability to undergo structural and functional changes in response to environmental stimuli, may be modulated by several stimuli. As such, changes in insulin signaling may affect brain plasticity and promote insulin resistance, for example [44]. It was found that insulin resistance is associated with enhanced brain glucose uptake during euglycemic hyperinsulinemia [161], as well as with lower arterial blood flow and reduced cortical perfusion in cognitively asymptomatic middle-aged adults [162]. Insulin resistance is associated with significantly lower cerebral glucose metabolism, which in turn, may predict worse memory performance [163]. Subjective cognitive complaints, changes in memory and brain function are also dependent on age [164]. 

As described earlier, insulin stimulates the transport of glucose into neurons by induction of GLUT 4 translocation to the neuron cell membrane. This movement is important during periods of high metabolic demand, such as during learning [40], and is known to enhance spatial memory [42]. Therefore, deregulation of insulin-dependent glucose transport in the brain may be a cause of cognitive impairment [40]. It is still not completely understood how insulin resistance affects the cognitive profile. Therefore, all of these suggestions need further investigation.

## 5. Therapy

Patients with AD demonstrate a deregulation in insulin signaling in specific brain areas, such as the hippocampus. Brain insulin resistance is commonly observed in early AD; as such, it has been proposed that therapy based on insulin sensitizer, diet or lifestyle, or insulin itself, may be a worthwhile approach. Certainly, as AD is defined as Type 3 diabetes mellitus, or brain diabetes mellitus, some of the proposed therapies are similar to that used in patients with diabetes mellitus.

### 5.1. Insulin Therapy

Peripheral insulin administration, a therapy used in the case of patients with diabetes, may induce hypoglycemia in AD patients who do not have diabetes; in addition, this treatment may be ineffective as insulin transport across the BBB is impaired. Therefore, studies have examined the potential for intranasal administration of insulin. This approach would allow the hormone to bypass the BBB by travelling to the brain via bulk flow along the trigeminal perivascular channels of the olfactory nerve [165]. Observations performed on cognitively healthy adult subjects indicate that intranasal insulin treatment affects various CNS measures, including MRI, EEG and MEG [18]. Intranasal administration of insulin (acutely and for 21 days) improves episodic memory in patients with mild cognitive impairment (MCI) or AD [166] and modulates beta-amyloid accumulation in early AD [167]. 

Studies have found 21-day treatment with low dose (20 IU) or high dose (40 IU) of long-lasting insulin detemir in 60 adults with MCI or AD resulted in improved memory, but only in the high-dose group [168]. In addition, 120-day administration of 20 IU or 40 IU insulin in 104 participants showed that both doses improved performance on the Alzheimer’s Disease Assessment Scale-Cognitive subscale (ADAS-Cog12), a measure of global cognition [169]. In other clinical studies, positive results demonstrated that intranasal insulin improves memory impairments in AD and in mild cognitive memory patients; the authors also note that the APOE ε4 allele does not appear to have any effect on insulin treatment efficacy [170]. Insulin treatment has been found to improve recall of verbal information after a delay, as well as other cognitive processes, such as orientation, judgment, social interactions and home activities, as rated by care givers [167]. This therapy also improves the CSF Aβ 40/42 and Aβ 42 to Tau ratios, biomarkers of AD [167,171], which suggests improvement. More information on intranasal insulin therapy is available in other studies [124,165,172,173].

### 5.2. Insulin Sensitizers Therapy

The use of insulin sensitizers is a strategy to make tissue more responsive to lower insulin concentrations. Although several insulin sensitizing compounds have been tested to restore CNS insulin sensitivity, information regarding their effects on the human brain are scarce [165]. The results of a small pilot study in humans may suggest that the peroxisome proliferator-activated receptor (PPAR-γ) agonists may preserve or improve cognitive function in AD [174], due to their possible positive effect on insulin signaling in the brain. One group of 30 patients with MCI and AD demonstrated improved memory and stabilized plasma Aβ_42_ concentration after six months of treatment with rosiglitazone [175]; in addition, a phase 2 trial revealed cognitive improvement after 24 weeks of treatment with rosiglitazone in only *APOE ε4* non-carriers [176], whereas a phase 3 trial showed no effects on cognition, regardless of *APOE* genotype [177]. 

Currently the evidence that insulin sensitizers may enhance insulin sensitivity in the brain is quite weak, and it is unclear whether they may act as effective therapeutic agents in AD. As such, this therapy needs further investigation.

### 5.3. Insulin Stimulating/Releasing Hormones Therapy

Insulin is not an ideal drug as a major treatment for AD, as higher insulin levels cause insulin desensitization. One promising alternative approach is the therapeutic administration of incretins, an example of which is glucagon-like peptide (GLP-1), an insulinotropic peptide generated by cleavage of proglucagon and secreted by small intestinal L cells following food intake. Administration stimulates neuritic growth in the CNS and plays a neuroprotective role against glutamate-mediated exo-toxicity, oxidative stress and cell death. GLP-1 can cross the BBB and can reduce the brain Aβ burden in AD [178]. However, as it has a short half-life of only a few minutes, its practical use for long-term therapy is limited. Therefore, synthetic longer-lasting analogues of GLP-1, such as the GLP-1 receptor agonist liraglutide, have been designed. Unfortunately, little information is available on their therapeutic effects in AD. Studies performed on small groups of patients indicate that subcutaneous liraglutide prevents brain glucose consumption, but does not appear to have any effect on Aβ load or cognition [179]. In animal studies, liraglutide treatment has been found to prevent memory impairment, loss of synapses and deterioration of synaptic plasticity, and to reduce Aβ plaque and soluble oligomer burden and inflammation. It has also been reported to increase neurogenesis in the hippocampal formation. However, no therapeutic effects have yet been demonstrated in humans [107]. Larger clinical trials examining the neuroprotective effects of liraglutide in AD are currently underway [173]. 

Glucose-dependent insulinotropic polypeptide (GIP), like GLP-1, is rapidly degraded by dipeptidyl peptidase-4 (DPP-4) following secretion by cells of the gastrointestinal tract. Its long-lasting analogue, D-Ala(2)GIP, was found to protect memory formation and synaptic plasticity, normalize stem cell proliferation, reduce Aβ_42_ plaques, and activate microglia and astrocytes in a mouse model of AD [180]. Despite this, its role as a therapeutic agent in AD is unknown and remains under investigation.

### 5.4. Other Antidiabetic Drug Therapies

Other antidiabetic drugs are also under investigation in AD therapy. Biguanides, such as metformin, are oral hypoglycemic drugs. Experimental studies indicate that this may have neuroprotective properties, manifested as preventing etoposide-induced apoptotic cell death in primary neurons and improving oxygen-glucose in neuronal injury [95]. Although metformin activity does not appear to be significantly related to the presence of ApoE ε4 or depression, long-term treatment (>6 years) has been found to reduce the risk of cognitive decline [95]. In subjects aged 50 years and older, metformin was found to significantly decrease the risk of dementia as compared to a non-medication group [181]. Hence, metformin may have possible confounding effects in the management of AD/neurological function and cognition, and additional research, including clinical studies, is needed. 

Another potential therapeutic option is the use of sulfonylureas, such as glyburide and glipizide, which inhibit mTOR activation. Glyburide inhibits the inflammasomes responsible for the elevation of proinflammatory cytokines known to cause neuroinflammation associated with AD [182]. However, clinical studies have not found these drugs to alter the risk of developing AD [181]. Several other antidiabetic and non-antidiabetic drugs have been proposed as therapeutics in AD [95,122], as have apomorphine [183], curcumin [184] and somatostatin [185].

### 5.5. Non-Pharmacological Approaches

Several studies performed on animal models indicate that a high-fat diet impairs brain insulin signaling. High-fat and high-glycemic diets lower insulin concentration in the CSF, and this is correlated with changes in CSF Aβ_42_. A low-fat and low-glycemic diet increases insulin concentrations in individuals with MCI to similar levels found in cognitively healthy people [186]. In addition, AD patients with omega-3-fatty acid deficiencies are more likely to demonstrate aging-associated cognitive impairment; in such cases, dietary supplementation with these fatty acids yields neuroprotective effects, manifested as a reduced risk of cognitive impairment in AD [187]. Eicosapentaenoic acid (EPA) and docosahexaenoic (DHA) reduce Aβ fragment formation, and DHA enhances synapse formation [39,188].

An active physical lifestyle can also protect against cognitive impairment in the elderly [189]. Physical activity reduces the risk of Alzheimer’s disease [190]. Animal studies showed that exercise improves brain insulin sensitivity, enhances mitochondrial function, reduces oxidative stress, and reduces Tau hyperphosphorylation and aggregation in neurons [191,192]. However, no human studies have been performed on the effect of exercise on brain insulin sensitivity, only on peripheral insulin sensitivity [107,165].

## 6. Summary

Brain insulin resistance plays an important role in development and progress of Alzheimer’s disease; most significantly, it increases oxidative stress and stimulates Aβ_42_ production and Tau protein phosphorylation. These in turn impair mitochondrial function, cognitive function and memory. In addition, several other pathologies are also known to be associated with AD. As AD is regarded as diabetes mellitus of the brain, or as type 3 diabetes mellitus, several therapies used to treat AD patients are based on those used for diabetic patients. Unfortunately, relatively little is currently known about these important treatments, and further investigations, clinical studies in particular, are needed to clarify their effects.

## Figures and Tables

**Figure 1 ijms-22-09987-f001:**
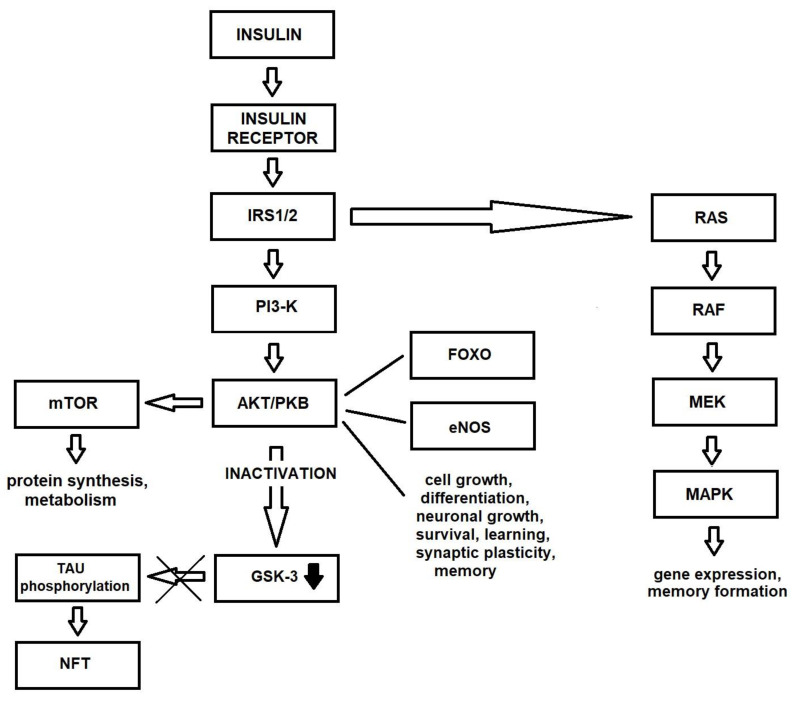
Insulin signaling pathway in healthy brain.

**Figure 2 ijms-22-09987-f002:**
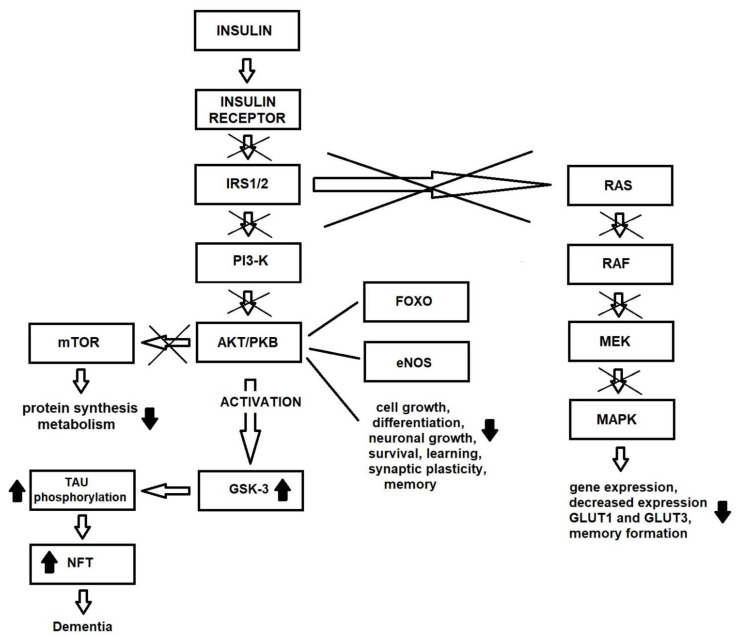
Changes in insulin signaling pathway due to insulin resistance in brain.

## Data Availability

Not applicable.

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
