# Peer review of "Insulin and Insulin Resistance in Alzheimer’s Disease"

_ijms, 2021, doi:10.3390/ijms22189987_

Round 1

Reviewer 1 Report

Review of a manuscript “Insulin and insulin resistance in Alzheimer’s disease” by Sędzikowska and Szablewski submitted to IJMS.

Alzheimer’s disease is the most prevalent neurodegenerative disorder bringing enormous harm to patients, their relatives and to the health care system. Furthermore, patients with Alzheimer’s disease often have comorbidity with other disorders, including diabetes mellitus, type 2. Investigation of molecular and cellular mechanisms of Alzheimer’s disease and its association with type 2 diabetes is very important biomedical issues. The authors summarized published data on the role of brain insulin resistance in Alzheimer’s disease pathogenesis, including the effect on oxidative stress, Aβ42 production and Tau phosphorylation. The topic of the review is essential and vital for the development of new diagnostic and treatment of both Alzheimer’s disease and type 2 diabetes. The review is well written and will be interesting for IJMS readers.

The following corrections and additions should be made:

Abstract:

1 “Insulin and insulin-like growth factor-1 receptors are expressed on all cell types in the central nervous system, and with widespread distribution in the brain confirm that insulin signaling plays important and diverse roles in this organ; it is known to regulate glucose metabolism, support cognition, enhance the outgrowth of neurons, modulate the release and uptake of catecholamine, and regulate the expression and localization of GABA”.

This sentence is too long and should be split into 2-3 sentences for better understanding and clarity. The fragment “…and with widespread distribution in the brain confirm that insulin signaling plays important…” should be rewritten to become clearer.

2 “…and affect plasticity possibly neurodegeneration.” This looks like unfinished sentence and needs clarification. May be “ and” is missing before plasticity?

Keywords: type 2 diabetes should be added

“Insulin signaling in the brain. Insulin and insulin-like growth factor 1 (IGF-1) exert biological effects through the insulin receptor (IR)”. This abbreviation has been already done above.

In 3.1. Insulin and brain glucose metabolism.

After the sentence:”This result suggests that the deregulation of insulin-de[1]pendent glucose transport in several brain regions may cause cognitive impairment [40].” The authors should add the following reference : “Caveolin: a new link between diabetes and Alzheimer’s disease. Cell Mol Neurobiol, 2020, 40 (7):1059-1066 doi: 10.1007/s10571-020-00796-4”

In 4.2. Insulin resistance and amyloid-β pathology

“Amyloid precursor protein (APP) is a transmembrane protein expressed in many tissues including the CNS, and is concentrated in the synapses of neurons. It functions as a cell surface receptor and has been implicated as a regulator of synapse formation, neural plasticity, antimicrobial activity and iron export.”

Underlying of several words in this sentence is not required.

Throughout the whole manuscript the authors should be consistent in using Alzheimer’s disease in full writing or abbreviated form.  

References. It is unclear why the names of certain authors in bibliography is underlined? This looks weird.

Author Response

Dear Reviewer

Thank you very much for your review, opinion and suggestions. According to your suggestions, we made changes as below:

1) This long sentence was divided into 3 separate sentences.

2) We think, that this sentence is correct, and "and" is necessary.

  • Keywords: type 2 diabetes mellitus is added.
  • Insulin signaling in brain. Repeated (IR) was excluded.

3.1. Suggested reference is added (192)

4. All underlines are excluded. We don't know what happen?

And again thank you very much, now article will be more easy and friendly for readers.

Reviewer 2 Report

ijms-1374409: INSULIN AND INSULIN RESISTANCE IN ALZHEIMER’S DISEASE
Special Issue: Pathogenesis of Alzheimer's Disease

This is a comprehensive review, covering almost all the topics about insulin and Alzheimer's disease (AD). The manuscript is well written and almost ready for publication. The following suggestions may help to increase the value of this review.
Although this is a matter of the Editorial Office, the manuscript PDF file provided for the review has no line numbers. It is very hard to pinpoint the lines mentioned below.

(1) The graphical summaries are recommended. At least two figures should be included. One explains the role of the insulin system in the physiological functions of the central nervous system (CNS). The other explains the pathological changes of the insulin system in AD. 

(2) The insulin production in CNS should be discussed. In the end of "1. Introduction", the function of insulin synthesized in CNS should be summarized (for example, as described in the introduction of 10.1172/jci.insight.131682).

(3) Page 1/28 in the introduction " Insulin can cross the blood-brain barrier (BBB) from the circulation to the brain via the capillary endothelial cells of the BBB by a selective, saturable, receptor-dependent mechanism [3]." The BBB is also affected somehow in dementia (for example, 10.1523/jneurosci.2506-18.2019). The change of BBB and insulin transport to CNS in AD should be briefly discussed somewhere (for example, before "5. Therapy").

(4) The relationship between the insulin system and the neurodegenerative diseases other than AD should be briefly summarized at the beginning of "4. Insulin resistance and Alzheimer’s disease" (for example, 10.1111/bph.14471 or 10.1038/s41531-016-0001-1). Whether or not insulin resistance accelerates dementia other than AD (for example, frontotemporal dementia, dementia with Lewy bodies, and so on) is an important point of view to understand the role of insulin in AD. 

(5) The cerebrovascular damages by insulin resistance in AD should be discussed. The dementia with AD pathology alone is very rare. Almost all the elderly patients of AD have some level of cerebrovascular pathologies as atherosclerosis. Insulin resistance should accelerate cerebrovascular damages. This matter should be discussed.

(6) Page 2/28 in 2. The role of insulin in <the central nervous system>
"On the other hand, changes of insulin signaling in <the central nervous system> may accelerate brain aging, affect plasticity and may be involved in the process of neurodegeneration [18]."
<the central nervous system> should be CNS.

(7) page 7/28 in 4.2. Insulin resistance and amyloid-beta pathology
in the <synapses> of <neurons>. It functions as a <cell surface receptor> and has been implicated as a regulator of synapse formation, <neural plasticity>, antimicrobial activity and <iron export>. APP
The underline of the parenthetical words above may be unnecessary.

End of File

Author Response

Dear Reviewer

Thank you very much for your review, opinion and suggestions. According to your suggestions, we made changes as below:

1) The graphical summaries is included.

2) The production of insulin in CNS is discussed in the end of "Introduction"

3) Suggested information is included, however in other place (In Introduction).

4) This suggested point is included at the "4".

5) We think, that manuscript is long, and therefore additional informations, however, important, will make mistakes.

6) "the central nervous system" is changed - CNS

7) The underline is excluded.

And again thank you very much. Now, article is more easy and friendly for readers.